# High PD-L1 Expression in HRS Cells and Macrophages in Tumor Immune Microenvironment Is Associated with Adverse Outcome and EBV Positivity in Classical Hodgkin Lymphoma

**DOI:** 10.3390/ijms26125592

**Published:** 2025-06-11

**Authors:** Antonija Miljak, Antonia Pavlović, Benjamin Benzon, Lučana Vicelić Čutura, Davor Galušić, Milan Vujčić, Viktor Blaslov, Merica Glavina Durdov

**Affiliations:** 1Department of Internal Medicine, Division of Haematology, University Hospital of Split, 21000 Split, Croatia; amiljak@kbsplit.hr (A.M.); lvicelic@kbsplit.hr (L.V.Č.); mivujcic@kbsplit.hr (M.V.); vblaslov@kbsplit.hr (V.B.); 2Department of Pathology, Forensic Medicine and Cytology, University Hospital of Split, 21000 Split, Croatia; apavlovic@kbsplit.hr (A.P.); mgladur@kbsplit.hr (M.G.D.); 3School of Medicine, University of Split, 21000 Split, Croatia

**Keywords:** Hodgkin lymphoma, EBV, PD-L1, HRS, macrophage, outcome

## Abstract

Programmed death-ligand 1 (PD-L1) on tumor cells, including Hodgkin and Reed–Sternberg (HRS) cells in classical Hodgkin’s lymphoma (cHL), suppresses immune responses in the tumor immune microenvironment (TME). We analyzed PD-L1 expression in macrophages and HRS cells of 98 cHL cases and correlated the findings with clinicopathological features, overall survival (OS), and progression-free survival (PFS). Epstein–Barr virus (EBV) was detected by in situ hybridization for EBV-encoded RNA. Ten high-power fields were evaluated to count the total number of macrophages and PD-L1+ macrophages, and to calculate PD-L1 histoscore (H-score) in HRS cells. EBV-positive cHL was found in 22.5% of patients. The median H-score was 80 (range 0–300). Bulky disease was associated with a lower number of PD-L1+ macrophages, and extranodal disease with a higher number (*p* = 0.05). EBV-positive cHL showed a higher PD-L1 H-score in HRS cells and a greater number of PD-L1^+^ macrophages (*p* = 0.005); both of these features, along with the proportion of PD-L1^+^ macrophages, were associated with shorter PFS and OS (*p* < 0.001). High PD-L1 expression in HRS and macrophages may be linked to worse clinical outcomes.

## 1. Introduction

The unusual features of cHL are its distinctive histological appearance, age bimodality, association with EBV, B symptoms, amplification of the gene that ensures PD-L1 expression, and unpredictable therapeutic response. Although most patients achieve complete remission (CR) and have a favorable prognosis, 10% to 30% of patients treated with standard therapy experience relapse or refractory (R/R) disease and have a poor long-term prognosis [1]. Possible reasons include, among others, the complexity of the TME and the interplay between immune evasion mechanisms and therapeutic interventions [2]. In some types of cancer, PD-L1 expression in tumor cells is induced by IFN-γ, which is secreted by surrounding T lymphocytes [3,4]. In cHL, PD-L1 is constitutively expressed due to amplification of 9p24.1 in HRS cells [5]. Blockade of the PD-1/PD-L1 axis with monoclonal antibodies targeting PD-1 (programmed cell death 1), known as immune checkpoint inhibitors (ICIs), has recently been successfully used in treating R/R cHL [6,7,8]. Therapeutic response appears to depend on the level of PD-L1 expression on HRS and macrophages [9]. The EBV-positive status of cHL impacts the TME pattern and the presence of regulatory cells [10]. Some studies show that EBV-positive cHL has higher PD-L1 expression on HRS, but the results are not unequivocal [11,12,13]. This study aimed to determine the expression of PD-L1 in the HRS and macrophages in the TME and to correlate them with EBV status, clinicopathological parameters, and outcome.

## 2. Results

The study cohort consisted of 98 cHL patients diagnosed and treated at University Hospital Split, Croatia, over 10 years. Patients ranged in age from 11 to 83 years, median 32.5, interquartile range (IQR) 21.8–50. There were 51 (52.1%) males and 47 (47.9%) females. Twenty-two (22.4%) patients had relapsed (n = 10) or experienced refractory disease (n = 12), and nine (41%) died due to disease progression; five patients died from other causes. EBV-positive cHL was confirmed by EBER in situ hybridization in 22 (22.5%) patients. PD-L1 expression in HRS and macrophages was analyzed by double immunohistochemistry, using PD-L1 and CD163 antibodies (Figure 1).

In 95/98 cases of cHL, malignant HRS cells were positive for PD-L1 to a certain extent and intensity. Instead of a qualitative assessment, a more precise H-score (yielding from 0 to 300) was calculated as the percentage of positively stained HRS multiplied by an intensity score, graded from weak (1+) to moderate (2+) to strong (3+). The absolute number and the proportion of PD-L1+ macrophages were counted manually. The results are shown in Table 1.

### 2.1. Association of PD-L1 Expression on HRS Cells with Clinical and Pathological Variables and Outcome of Classical Hodgkin Lymphoma

The median H-score for PD-L1 expression in HRS cells was 80 (IQR 36.5–142.8). Patients with EBV-positive cHL had approximately twice the H-score as those with EBV-negative cHL (*p* = 0.005) (Table 2). PD-L1 expression on HRS cells was not associated with the number of macrophages (*p* = 0.2329) and PD-L1+ macrophages (*p* = 0.3808), nor with the proportion of PD-L1+ macrophages (*p* = 0.7212). Pearson’s test showed no correlation of H-score values with age (*p* = 0.16) and hemoglobin level (*p* = 0.78).

Patients were divided into two groups using the median PD-L1 H-score as the cutoff value, and OS and PFS were analyzed (Figure 2).

Patients with a high H-score died 4.55 times more frequently (68% CI: 4.24–5.03) than patients with a low H-score (*p* < 0.0001, evidence ratio > 100) (Figure 1a). They also relapsed 1.38 times more frequently (68% CI: 1.1–1.43) compared with patients with a low H-score (Figure 1b). The plateau probability of PFS was 81.93 ± 0.2% in the low H-score group and 68.94 ± 0.17% in the high H-score group (*p* < 0.0001; evidence ratio > 100).

### 2.2. Association of PD-L1 Expression on Macrophages with Clinical and Pathological Variables and Outcome of Classical Hodgkin Lymphoma

The average number of macrophages per field at 1000× magnification was 6.8, of which 5.4 (86%) were PD-L1+ macrophages. The highest absolute number of macrophages was found in mixed cellularity subtype, of which 89.8% were PD-L1+ macrophages (*p* = 0.02). In nodular sclerosis, the number of macrophages was one-third lower, with 85.3% PD-L1+ macrophages (*p* = 0.0002) (Table 3).

A higher number of PD-L1+ macrophages correlated with extranodal disease (*p* = 0.05), whereas a lower number and proportion of PD-L1 + macrophages correlated with bulky tumor (both *p* = 0.04) (Figure 3).

Higher absolute numbers of macrophages (*p* = 0.008) and PD-L1+ macrophages count (*p* = 0.005) were found in EBV-positive than in EBV-negative cHL, but the proportion of PD-L1+ macrophages was not significantly different (*p* = 0.11). The proportion of PD-L1+ macrophages did not correlate with age (*p* = 0.22), hemoglobin level (*p* = 0.21), stage (*p* = 0.11), and B symptoms (*p* = 0.17).

According to the mean value of macrophages, PD-L1+ macrophages, and the proportion of PD-L1+ macrophages, two groups were formed and compared regarding survival (Figure 4).

Patients with high macrophage counts died 6 times faster (68% CI: 3.96–26) than patients with low counts (*p* < 0.0001, ER > 100) (Figure 4a). Patients with high PD-L1+ macrophage counts reached a survival probability plateau at 73.61 ± 1.7%, while patients with low counts had a plateau at 81.8 ± 1.2% (*p* < 0.0001, ER > 100) (Figure 4b). Patients with high proportions of PD-L1+ macrophages experienced death at more than a 100-fold higher rate compared with those with low proportions (*p* < 0.0001, ER > 100) (Figure 4c). Patients with low macrophage counts had a plateau PFS probability of 82.96 ± 0.16%, and patients with high counts had a plateau of 68.68 ± 0.21% (*p* < 0.0001, ER > 100) (Figure 4d). Patients with low counts of PD-L1+ macrophages had a plateau PFS probability of 82.96 ± 0.1%, and patients with high counts had a plateau of 68.66 ± 0.26% (*p* < 0.0001, ER > 100) (Figure 4e). Patients with high proportions of PD-L1+ macrophages had a plateau PFS probability of 65.8 ± 0.26%, and those with low counts had a plateau PFS probability of 85.38 ± 0.15% (*p* < 0.0001, ER > 100) (Figure 4f).

## 3. Discussion

In this single-institution study, high PD-L1 H-score expression in HRS cells adversely affected the clinical outcome of patients with cHL. Higher numbers and proportions of PD-L1+ macrophages were associated with shorter PFS and OS. A higher number of PD-L1+ macrophages correlated with extranodal disease, whereas their lower number and proportion correlated with bulky tumor. Furthermore, EBV-positive cHL status correlated with higher PD-L1 H-score in HRS cells. This is the first study to consider both the number and proportion of PD-L1+ macrophages, along with the PD-L1 H-score in HRS cells, using high magnification under immersion and examination of whole slides, with cell counts comparable to those used in other IHC studies.

Hollander et al. immunohistochemically analyzed checkpoint molecule expression on leukocytes in the TME of cHL, using tissue microarrays. They found different cutoffs for proportions of PD-1+, PD-L1+, and PD-L2+ leukocytes, and confirmed that high proportions are associated with inferior outcomes [14]. In a molecular study of cHL, Roemer et al. used fluorescent in situ hybridization to evaluate *PD-L1/PD-L2* alterations and found that 9p24.1 amplification correlated with shorter PFS and increased PD-L1 expression [15]. Paydas et al. found that PD-1 and PD-L1 expression alone was not associated with outcome, but their co-expression in HRS cells and TME had an independent adverse effect on OS and PFS. Cases were considered PD-L1-positive when more than 5% of HRS expressed PD-L1, regardless of staining intensity [11]. Koh et al. found that PD1, but not PD-L1 or PD-L2, had an independent adverse effect on OS [16].

According to Steidl et al., high numbers of macrophages in TME of cHL are unfavorable for prognosis and therapy, which was later confirmed in a meta-analysis by Guo et al. [17,18]. In our study, high numbers of macrophages were associated with shorter OS and PFS. More importantly, we found that a high proportion of PD-L1+ macrophages was associated with poor outcomes. Karihtala et al. used multiplex IHC and digital image analysis on tissue microarrays to characterize macrophages in cHL and found that a high proportion of PD-L1+ macrophages was associated with poorer survival [19]. Carey et al. presented a unique TME topology in cHL, using multiplex immunofluorescence. Abundant PD-L1+ macrophages physically co-localized with PD-L1+ HRS cells and were in contact with CD4+ T cells, a subset of which expressed PD-1 [20].

In our study, we frequently observed PD-L1+ macrophages clustering near HRS. We also noted PD-L1-positive membrane extensions from macrophages to neighboring HRS cells, likely representing the PD-L1/L2 protein trogocytosis process, as previously described by Kawashima et al. [21]. According to Bettadapur et al., trogocytosis is the rapid intercellular transfer of small protein-bearing membrane fragments that bud from the donor cell and are then either integrated into the surface of the acceptor cell or internalized [22]. Zeng et al. reported that PD-1+ T cells in the vicinity of PD-L1+ HRS cells or macrophages can acquire PD-L1 via trogocytosis, potentially suppressing newly infiltrated PD-1+ T cells or those located further away from PD-L1+ HRS cells or macrophages [23]. These observations suggest a role for intercellular membrane remodeling in shaping the local immune microenvironment. Stewart et al., using a multi-omics approach, found that enrichment of the inflammatory classical dendritic cell-monocyte–macrophage network was associated with early relapse after treatment [24].

In our study, 22.5% of patients had EBV-positive cHL, consistent with findings from other developed countries [25]. Among all variables analyzed, only EBV status was associated with higher PD-L1 expression on HRS cells. Additionally, it correlated with increased numbers of PD-L1+ macrophages and total macrophages. Moyano et al. observed that CD163+ macrophages in the tonsils during primary EBV infection showed increased PD-L1 expression [26]. Yu et al., in a meta-analysis of lymphoproliferative diseases, demonstrated higher PD-L1 expression in tumor cells of EBV-positive cases [27].

In cHL, the results are inconclusive. Roemer et al. showed that the distribution of 9p24.1 genetic alterations in patients with EBV-negative and EBV-positive cHL patients was similar, although EBV-positive cHLs were more likely to have high PD-L1 H-scores [15], consistent with our findings. Paydas et al. did not find a significant association between PD-1 and PD-L1 expression and EBER positivity in cHL [11].

In the studies by Antel et al. and Jimenez et al., conducted in cHL cohorts with high prevalence of EBV (pediatric cases in Argentina and a population with high HIV prevalence in South Africa), no association between EBV presence and PD-L1 expression was observed [12,13]. Green et al., using IHC and genomic analysis, demonstrated that EBV in HRS cells may serve as an alternative mechanism for PD-L1 induction, even in cHLs with diploid 9p24.1 [28].

In our study, the most common subtype was nodular sclerosis, consistent with findings from other developed countries. Despite histological differences, PD-L1 expression on macrophages and HRS cells was not associated with the subtype. Current research also does not demonstrate a definitive association between PD-L1 expression and cHL subtypes.

In R/R cHL, standard treatment achieves disease control in only half of patients [29]. Roemer et al. found that ICI therapy in these patients can improve PFS when they exhibit high-level 9p24.1 copy gains and elevated PD-L1 expression on HRS cells [30]. NIVAHL and KEY-NOTE-C11 trials demonstrated that ICI therapy, with or without chemotherapy, can yield high complete remission rates and improved responses both in first-line [31,32] and R/R settings [6,7].

Reinke et al. reported a notable finding: ICI administration in the first-line setting led to the absence of HRS cells, as well as the depletion of regulatory T cells and PD-L1+ macrophages in TME only a few days after initiation of therapy [33]. These findings suggest that the TME structure may collapse with the disappearance of HRS cells. The mechanism of ICI action may therefore involve the withdrawal of survival-supporting factors rather than the activation of an immune response [33]. Consequently, incorporating PD-1 blockade into first-line therapy could enhance treatment efficacy and potentially reduce dependence on traditional chemotherapy and radiotherapy for selected patients.

The limitations of this study include a relatively small sample size, a low number of events, and the use of two histological methods rather than more advanced techniques. However, strengths include the precise assessment of PD-L1 expression on HRS cells using the H-score and microscopic analysis at high magnification under immersion, allowing a detailed evaluation of spatial relationships between HRS cells and macrophages.

## 4. Materials and Methods

### 4.1. Patient Characteristics

This retrospective study examined immunohistochemical data from tumor tissue in 98 patients diagnosed with cHL at University Hospital Split, Croatia, between 1 January 2008 and 31 December 2018. Clinical and demographic information, including modified Lugano stage, presence of extranodal disease, treatment history, survival status (alive/dead), progression-free survival (PFS), and overall survival (OS) in months, were obtained from the Department of Internal Medicine and the Department of Pediatrics. Paraffin blocks of tumor tissue were retrieved from the Clinical Department of Pathology, Forensic Medicine, and Cytology archives. The study protocol was approved by the Ethics Committee of University Hospital Split (2181-147-01/06/M.S.-19-2), and all procedures were performed following the Declaration of Helsinki.

Of the 112 patients diagnosed with cHL during this period, 98 were included in the study based on complete clinical follow-up data and matching tumor tissue (Appendix A). The end of the follow-up was on 20 October 2023. The median follow-up duration was 97 months (range: 0.3–202 months). Relapse was defined as disease recurrence after completion of treatment, with confirmation of complete remission by MSCT or PET-CT.

Patients were treated according to CROHEM (Croatian Cooperative for Hematological Diseases) and NCCN (National Comprehensive Cancer Network) guidelines. The primary treatment regimen was ABVD (doxorubicin, bleomycin, vinblastine, and dacarbazine), followed by radiotherapy when indicated—usually in patients with limited disease and cases of bulky tumor, slow tumor regression, or localized residual mass in initially disease. BEACOPP (bleomycin, etoposide, doxorubicin, cyclophosphamide, vincristine, procarbazine, and prednisone) was administered since January 2017 for five patients with advanced diseases. Pediatric patients (n = 6) received OEPA (vincristine advanced, etoposide, prednisone, and doxorubicin) followed by COPP (cyclophosphamide, vincristine, procarbazine, and prednisone). Additionally, two patients older than 65 years received four cycles of COPP. Patients with R/R cHL (n = 22) were mainly treated with high-dose salvage chemotherapy, including DHAP (cisplatin, cytarabine, and dexamethasone) or ICE (ifosfamide, carboplatin, and etoposide), followed by autologous stem cell transplantation.

### 4.2. Methods

Approximately 3 μm thick slides were cut from paraffin blocks of tumor tissue and mounted on silanized glass. Presence of EBV in tumor cells was analyzed by EBER in situ hybridization, using the Ventana Ultra Benchmark System with an EBER oligoprobe (INFORM EBER Probe, Ventana Roche Diagnostics, Basel, Switzerland) and the ISH iView Blue Detection Kit (Ventana Roche Diagnostics). A positive reaction was defined as black-blue nuclear staining in at least one HRS cell. The positive external control used was EBV-positive nasopharyngeal carcinoma.

Double immunohistochemical (IHC) staining was performed using mouse monoclonal antibodies to CD163 (clone MRQ26, Cell Marque, Darmstadt, Germany) and PD-L1 (263 Assay, Ventana, Roche, Basel, Switzerland) on the Ventana Ultra Benchmark (Ventana Medical Systems, Tucson, AZ, USA). The secondary antibodies used were DAB/IHC and Fast Red/IHC (Ventana, Roche, Switzerland). A positive reaction was defined as red membrane/cytoplasmic staining for CD163 and brown membrane staining for PD-L1.

Slides were analyzed using an Olympus 41 BX light microscope (Olympus, Tokyo, Japan). Each slide was examined at 200× and 400× magnification to assess tumor tissue, remove areas of technical artifacts, and identify several representative regions for analysis. Selected areas were then analyzed at 1000× magnification in immersion oil. Ten nonoverlapping fields were photographed, and photomicrographs were analyzed using CellSens Standard 1.9 imaging software (Olympus, Japan). Single- and double-positive cells were counted on the photomicrographs, and two experienced pathologists independently scored the values for HRS cells and macrophages.

#### 4.2.1. Analysis of PD-L1 in HRS Cells

The average number of HRS cells in 10 analyzed fields of each cHL case was 40. PD-L1 quantification on HRS cells followed the method described by Roemer et al. [15]. All detected HRS cells were analyzed, and the intensity of PD-L1 expression was assessed (0 = no staining, 1 = weak staining, 2 = moderate staining, 3 = strong staining). The number of HRS with positive staining (1 to 3+) was determined, and the percentage of positive cells was calculated. Finally, staining intensity was multiplied by the percentage of cells according to formula [1 × (% cells 1+) + 2 × (% cells 2+) + 3 × (% cells 3+)], resulting in a H-score ranging from 0 to 300.

#### 4.2.2. Analysis of PD-L1 in Macrophages

CD163+ macrophages were analyzed in 10 fields, and the average number per field was calculated. Macrophages were classified as PD-L1-positive if they expressed PD-L1 at any intensity. The ratio of PD-L1+ macrophages to the absolute number of macrophages was then calculated.

### 4.3. Statistical Analysis

Continuous variables are presented as median and interquartile range (IQR), and categorical variables as fractions (i.e., percentages). Differences in continuous variables between groups were modeled using ANOVA or *t*-test. Correlations were quantified using Kendall’s tau. Survival curves were estimated using the Kaplan–Meier method. Survival curves (i.e., time-to-event data) were modeled parametrically using an exponential decay-to-plateau model. Additionally, multivariate analysis of time-to-event data was performed using a discrete-time model (i.e., logistic regression), where the hazard for the event was used as the dependent variable (Appendix A). Model comparisons were performed using AIC-based evidence ratio (ER), F-test, and standard model diagnostic tools (R², residuals, partial residual plots, etc.). Uncertainty in parameter estimates is expressed as standard error (i.e., 68% confidence interval) or 85% confidence interval. *p*-values were interpreted according to the American Statistical Association statement on *p*-values. Statistics were performed on GraphPad Prism 11.0 (GraphPad Software, Boston, MA, USA).

## 5. Conclusions

Integrated assessment of PD-L1 expression on HRS cells, proportion of PD-L1+ macrophages, and EBV status in the first biopsy of cHL could signal the risk of adverse outcomes and the eventual benefit of using checkpoint inhibitors in first-line treatment, but large-scale studies are needed.

## Figures and Tables

**Figure 1 ijms-26-05592-f001:**
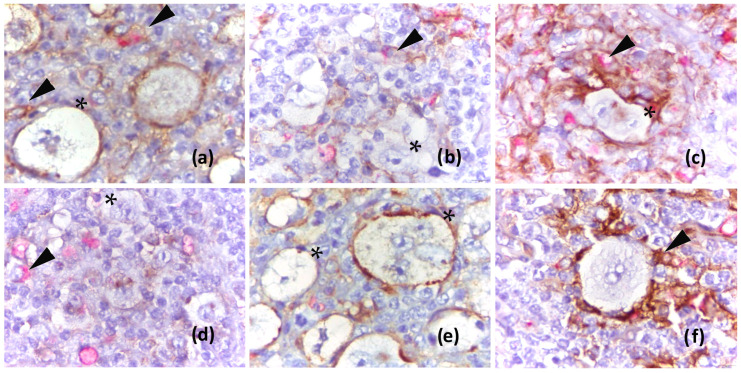
Tumor microenvironment in classical Hodgkin lymphoma immunostained for PD-L1 (brown) and macrophage marker CD 163 (red). (**a**) PD-L1-positive HRS cells (*) with a few PD-L1+ macrophages in the vicinity (arrow); (**b**) PD-L1-negative HRS cells and predominantly PD-L1+ macrophages; (**c**) PD-L1-positive HRS cell surrounded by numerous PD-L1+ macrophages; (**d**) PD-L1-negative HRS cells and PD-L1-negative macrophages; (**e**) variable PD-L1 positivity in HRS cells, highlighting the importance of measuring the H-score; (**f**) PD-L1 extensions from PD-L1+ macrophages to HRS cells are indicative of trogocytosis, the rapid intercellular transfer of membrane fragments between cells; (all at 1000× magnification).

**Figure 2 ijms-26-05592-f002:**
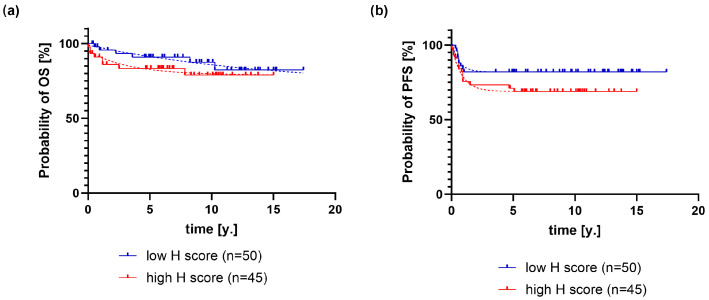
Clinical outcome of patients with cHL stratified by low (<80) and high (>80) PD-L1 H-scores in HRS cells. Kaplan–Meier curves for (**a**) overall survival (OS) and (**b**) progression-free survival (PS) with exponential model (dashed line). R2 for all curves is greater than 90%. Legend: y.—years.

**Figure 3 ijms-26-05592-f003:**
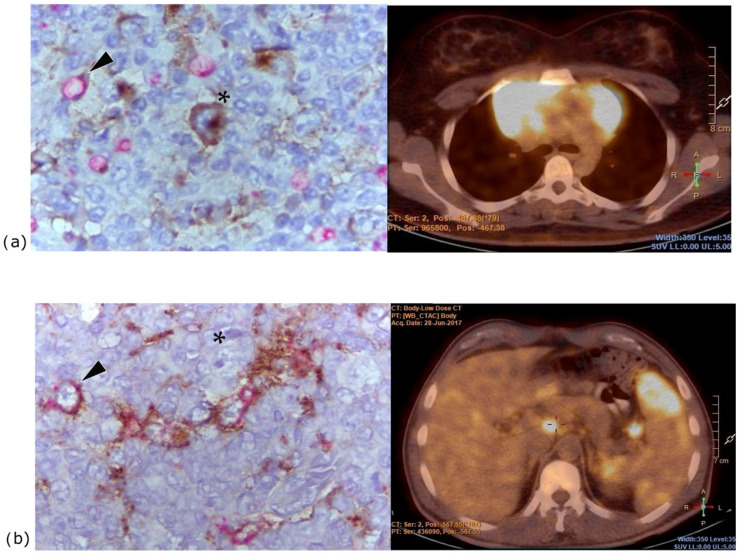
PD-L1 expression on macrophages in TME and PET/CT findings. (**a**) Low numbers of PD-L1-negative macrophages correlated with high tumor mass. PD-L1-negative macrophages were predominantly found in the TME in this patient with bulky mediastinal disease on PET/CT. (**b**) High numbers of PD-L1+ macrophages were associated with extranodal disease. In this patient, who had spleen, liver, and left lower lobe involvement, macrophages were predominantly positive for PD-L1. (Photomicrographs: Magnification 1000×, PD-L1 red and CD163 brown staining. HRS cells are indicated by asterisks and macrophages by arrows.)

**Figure 4 ijms-26-05592-f004:**
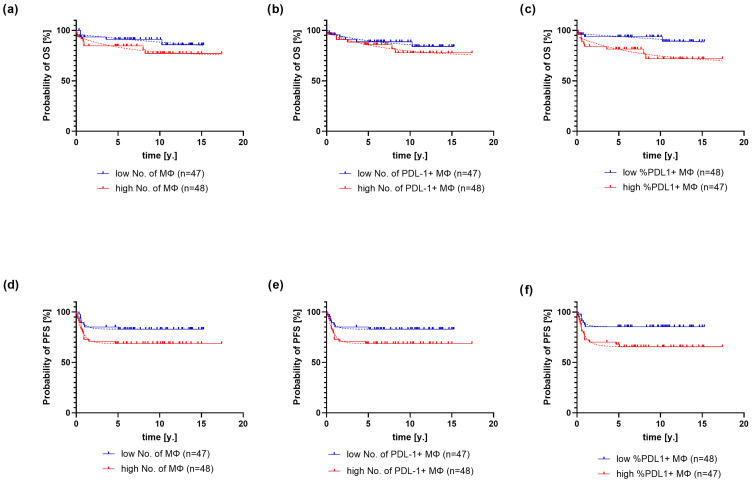
OS (overall survival) and PFS (progression-free survival) of patients with cHL, stratified by absolute macrophage count (**a**,**d**), as well as absolute number (**b**,**e**), and proportion of PD-L1+ macrophages (**c**,**f**). Legend: MΦ—macrophages, y.—year.

**Table 1 ijms-26-05592-t001:** Expression of PD-L1 in HRS and macrophages in classical Hodgkin lymphoma (n = 98).

Variables	Median, IQR
PD-L1+ HRS H-score	80 (36.5–142.8)
Macrophages	6.8 (5.3–8.9)
PD-L1+ macrophages	5.4 (3.6–7.1)
Proportion of PD-L1+ macrophages (%)	85.9 (65.4–91.9)

Legend: IQR—interquartile range.

**Table 2 ijms-26-05592-t002:** Median H-score for PD-L1 in HRS cells according to analyzed variables.

Variable	PD-L1 H-Score	IQR of H-Score	*p*
Histological subtype			0.10
Nodular sclerosis (n = 73)	83	40–50	
Mixed cellularity (n = 21)	75	31–121	
Lymphocyte-rich (n = 3)	20	3–80	
Lymphocyte-depleted (n = 1)	216	216–216	
Clinical stage			0.28
Early favorable (n = 23)	81	25–145	
Early unfavorable (n = 29)	67	35–118	
Advanced (n = 45)	85	40–190	
Extranodal disease			0.35
no (n = 66)	84	40–154	
yes (n = 30)	77	33–142	
Bulky disease			0.69
no (n = 72)	82	28–133	
yes (n = 26)	80	38–145	
B symptoms			0.42
no	75	36–143.5	
yes	85	37.5–143.5	
EBV status			0.005
no (n = 75)	70	34–104	
yes (n = 22)	136	82.5–182.3	

ANOVA Legend: IQR—interquartile range.

**Table 3 ijms-26-05592-t003:** PD-L1 expression in macrophages and their correlation with clinical and pathological variables in classical Hodgkin lymphoma.

Variable	Macrophages		PD-L1+ Macrophages		Proportion of PD-L1+ Macrophages			
Subtype	N	IQR	N	IQR	%	IQR		
Nodular sclerosis	6.3	4.65–7.73	4.9	3.21–6.65	85.3	63.3–91.2		
Mixed cellularity		9.6	7.62–10.65	8.0		6.4–10.31	89.8	83.9–93.9
Lymphocyte-rich	5.2	4.3–6.7	4.2	1.4–4.6	68.7	32.6–80.8		
Lymphocyte-depleted	8.7	8.7–8.7	4.6	4.6–4.6	52.9	52.9–52.9		
*p*	0.0002		<0.0001		0.02			
Extranodal								
no	6.65	4.92–7.92	5.0	3.21–6.8	85.1	63.1–90.2		
yes		7.48	5.3–9.8	6.3		4.0–8.42	88.7	76.7–94.9
*p*	0.18		0.05		0.06			
Bulky								
no	7.35	5.3–9.2	5.75	4.0–8.0	87.1	74.4–92.0		
yes		6.2	4.1–7.78	3.9		2.87–6.4	82.3	56.2–91.7
*p*	*0.07*		0.04		0.04			
EBV status								
negative	6.3	4.6–7.8	4.8	3.3–6.8	84	64.1–90.9		
positive	8.3	6.9–9.8	6.8	5.6–8.9	90.7	82.2 –94.6		
*p*	0.008		0.005		0.11			

ANOVA test. Legend: IQR—interquartile range.

## Data Availability

The data presented in this study are available on request from the corresponding author.

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
