# Peer review of "High PD-L1 Expression in HRS Cells and Macrophages in Tumor Immune Microenvironment Is Associated with Adverse Outcome and EBV Positivity in Classical Hodgkin Lymphoma"

_ijms, 2025, doi:10.3390/ijms26125592_

Round 1
Reviewer 1 Report
Comments and Suggestions for Authors
The data is both interesting and well-documented. The presentation could have been more concise. The removal of 2-3 pages will result in a more compact text that may be more easily appreciated.
I must admit that I am not fully versed in the rationale of interjecting "materials and methods" between discussion and conclusions. A more common strategy would be to place them after the introduction.
The paper's weakness lies in the group size, which has the potential to create an imbalance between high- and low-risk cases in the "high and low" H score cohorts.
Author Response
- Q: The presentation could have been more concise. The removal of 2-3 pages will result in a more compact text that may be more easily appreciated.
A: You are deffinitely right! We managed to shorten it, and I really hope it’s more readable now.
- Q „I must admit that I am not fully versed in the rationale of interjecting "materials and methods" between discussion and conclusions.“
A: I completely agree with you but we needed to follow the subtitle structure from the IJMS template, so we decided to keep it as is.
- Q: The paper's weakness lies in the group size, which has the potential to create an imbalance between high- and low-risk cases in the "high and low" H score cohorts.
A: Our sample size is close to the average reported in similar studies and reflects a long observation period within a single medical institution.
Reviewer 2 Report
Comments and Suggestions for Authors
General Comments:
The authors investigated PD-L1 expression on Hodgkin/Reed-Sternberg (HRS) cells and macrophages in diagnostic Hodgkin lymphoma (HL) specimens, analyzing associations with patient characteristics and clinical outcomes. While the limitations of immunohistochemistry (IHC) are appropriately acknowledged, particularly when compared to more comprehensive molecular techniques, the findings still contribute meaningfully to the existing body of knowledge on this topic.
The study includes a cohort of 98 patients diagnosed over a 10-year period. PD-L1 expression on HRS cells was quantified using an H-score, while expression on macrophages was assessed via both percentage of positive cells and absolute cell counts.
The manuscript is lengthy and would benefit from condensing. The authors might consider reformatting the work as a short report, emphasizing the key findings and transferring detailed tables and secondary analyses to the supplementary materials.
Specific Comments:
-
H-score Methodology:
Please clarify whether the H-score used for PD-L1 quantification on HRS cells follows the method described by Roemer et al. (2018). If so, this should be explicitly stated and referenced in the Methods section. -
Treatment Details:
The description of first-line treatment regimens is insufficient. While the majority of patients were reportedly treated with ABVD, more detailed information on the exact regimens, including any variations or risk-adapted approaches, should be provided. -
Terminology:
Consider using the more widely accepted abbreviation "TME" for the tumor microenvironment, rather than "TIM", which may cause confusion with "TIM" (T-cell immunoglobulin and mucin-domain containing) checkpoint molecules. -
Abstract (line 23):
Please include the full range of the H-score values in the abstract to better contextualize the findings. -
Survival Data Presentation:
The presentation of survival results is overly detailed and detracts from the main findings. Consider focusing only on significant or clinically relevant results, and streamlining the discussion accordingly. -
Cox Regression Terminology:
The phrase "fold-faster" used to describe hazard ratios is misleading. Cox regression hazard ratios reflect the relative risk of event occurrence, not a rate or speed. Please revise this terminology. -
Modelling Concerns:
The modeling approach applied to survival curves appears unconventional. The observed plateau in the survival curves at around one year is atypical for Hodgkin lymphoma, which usually has longer periods of progression-free survival. Could this be an artifact of the chosen modeling technique? -
Time Since Diagnosis (line 179):
Including "time since diagnosis" as a variable in the survival analysis is questionable. Since relapse risk is inherently time-dependent, this inclusion may introduce bias. Patients with longer follow-up who have not relapsed will naturally appear to be at lower risk.
Author Response
- Q: Please clarify whether the H-score used for PD-L1 quantification on HRS cells follows the method described by Roemer et al. (2018). If so, this should be explicitly stated and referenced in the Methods section.
A: Thank you for your suggestion! We used a method similar to that of Roemer et al. and now described it in the Methods and cited in the References.
- Q: Treatment Details: The description of first-line treatment regimens is insufficient. While the majority of patients were reportedly treated with ABVD, more detailed information on the exact regimens, including any variations or risk-adapted approaches, should be provided.
A: Thank you for your suggestion! We have now provided a more detailed description of the therapy. The great majority of patients were treated with the ABVD protocol during that period. In Croatia, we began using BEACOPP from January 2017, only for younger patients with advanced-stage disease, so only five patients received this treatment.
- Q: Consider using the more widely accepted abbreviation "TME" for the tumor microenvironment, rather than "TIM", which may cause confusion with "TIM" (T-cell immunoglobulin and mucin-domain containing) checkpoint molecules.
A: Thank you for your helpful suggestion. We have now corrected the abbreviation throughout the entire text.
- Q: Please include the full range of the H-score values in the abstract to better contextualize the findings.
A: Thanks for your observation. We added the full range to the abstract now.
- Q: Survival Data Presentation:
The presentation of survival results is overly detailed and detracts from the main findings. Consider focusing only on significant or clinically relevant results, and streamlining the discussion accordingly.
A: We've shortened the description of survival data and emphasized the significant results in the Discussion.
- Q: Cox Regression Terminology:
The phrase "fold-faster" used to describe hazard ratios is misleading. Cox regression hazard ratios reflect the relative risk of event occurrence, not a rate or speed. Please revise this terminology.
A: We did not apply the Cox proportional hazards regression for the analysis of survival data. Instead, we opted for a parametric survival model—specifically, the exponential decay model. Unlike Cox regression, which makes no assumptions about the shape of the survival function and may therefore lose statistical power, the exponential model offers a defined structure. It includes two key parameters: the rate constant, which allows for calculation of the population half-life (i.e., the event occurrence rate), and the plateau parameter—sometimes referred to as the “tail” of the survival curve—which reflects the proportion of the population in which events no longer occur.
Accordingly, we did not use hazard ratios or Cox regression in our analysis, and as such, there is no corresponding terminology to revise.
Reference: Harrell, F.E. (2015). Parametric Survival Models. In: Regression Modeling Strategies. Springer Series in Statistics. Springer.
- Q: Modelling Concerns:
The modeling approach applied to survival curves appears unconventional. The observed plateau in the survival curves at around one year is atypical for Hodgkin lymphoma, which usually has longer periods of progression-free survival. Could this be an artifact of the chosen modeling technique?
A: In our cohort, most relapses occurred early, predominantly within the first year after diagnosis, while late relapses were infrequent. The survival curves were first constructed from raw data using the Kaplan-Meier method, prior to any modeling. These initial curves already displayed long tails, i.e., a plateau, indicating that a subset of patients remained progression-free beyond the first year. Therefore, the observed plateau is reflective of the actual clinical data and not an artifact of the modeling technique.
- Q: Time Since Diagnosis (line 179):
Including "time since diagnosis" as a variable in the survival analysis is questionable. Since relapse risk is inherently time-dependent, this inclusion may introduce bias. Patients with longer follow-up who have not relapsed will naturally appear to be at lower risk.
A: It’s important to recognize that time is an essential component of any survival analysis. Whether using nonparametric methods (Kaplan–Meier, Nelson–Aalen), semiparametric models (Cox regression), parametric models, or discrete-time approaches, all analyses fundamentally depend on incorporating time to appropriately model time-to-event data. Without including time, conducting a meaningful survival analysis isn’t possible.
Reviewer 3 Report
Comments and Suggestions for Authors
I have carefully considered your manuscript. While the study addresses an important topic and presents interesting univariate associations, I regret to inform you that the manuscript lacks the in-depth analyses necessary to support the robustness of the findings.
The primary concerns that prevent acceptance are related to the statistical power and the robustness of the multivariate analysis, limitations that the authors themselves have acknowledged.
Specifically, the study's foundation is built upon a relatively small sample size of 98 patients, coupled with a low number of observed events (22 relapsed/refractory cases and 14 deaths by the end of the follow-up period). This limitation has resulted in insufficient statistical power, particularly impacting the ability to detect associations in the multivariate analysis.
Consequently, the multivariate modeling of both overall survival (OS) and progression-free survival (PFS) showed that adding the PD-L1 H-score and the percentage of PD-L1+ macrophages did not decisively improve the predictive model already based on established clinicopathological variables. The authors attribute this to the correlation between the new PD-L1 variables and existing predictors like EBV status and time since diagnosis, suggesting the PD-L1 markers may not add substantial independent information in this model.
Although the univariate analyses provide valuable initial insights, the lack of robust findings from the multivariate analysis due to the power limitations significantly weakens the conclusions regarding the independent prognostic value of the studied PD-L1 markers.
Moreover, the analysis is based solely on immunohistochemistry data, and the absence of deeper molecular or functional investigations further limits the interpretability and generalizability of the findings.
In Figure 4, the authors could have added arrows to more clearly indicate which cells are HRS cells and which are PD-L1-positive macrophages. Similarly, Figure 2 also lacks arrows, which would help in better identifying the macrophages.
Given these significant limitations impacting the ability to draw definitive conclusions, the manuscript appears too weak in its current form.
Author Response
- I have carefully considered your manuscript. While the study addresses an important topic and presents interesting univariate associations, I regret to inform you that the manuscript lacks the in-depth analyses necessary to support the robustness of the findings. The primary concerns that prevent acceptance are related to the statistical power and the robustness of the multivariate analysis, limitations that the authors themselves have acknowledged. Specifically, the study's foundation is built upon a relatively small sample size of 98 patients, coupled with a low number of observed events (22 relapsed/refractory cases and 14 deaths by the end of the follow-up period). This limitation has resulted in insufficient statistical power, particularly impacting the ability to detect associations in the multivariate analysis. Consequently, the multivariate modeling of both overall survival (OS) and progression-free survival (PFS) showed that adding the PD-L1 H-score and the percentage of PD-L1+ macrophages did not decisively improve the predictive model already based on established clinicopathological variables. The authors attribute this to the correlation between the new PD-L1 variables and existing predictors like EBV status and time since diagnosis, suggesting the PD-L1 markers may not add substantial independent information in this model. Although the univariate analyses provide valuable initial insights, the lack of robust findings from the multivariate analysis due to the power limitations significantly weakens the conclusions regarding the independent prognostic value of the studied PD-L1 markers.
A: Thank you for your comment. This analysis is based on data from a single medical institution, using a sample size that we believe is sufficient to draw meaningful conclusions from the evaluated histological parameters. Additionally, several other studies have also not demonstrated a significant association of PD-L1 expression in multivariate analysis. However, this does not undermine the relevance of our findings, as the primary purpose of analyzing PD-L1 expression is its potential therapeutic implication—particularly in the context of immunotherapy—as clearly discussed in our manuscript, rather than its standalone prognostic value. In addition, if our study is published then it could be used as a part of some future metanalysis in which such power issues would be mitigated.
- Q: Moreover, the analysis is based solely on immunohistochemistry data, and the absence of deeper molecular or functional investigations further limits the interpretability and generalizability of the findings.
A: Thank you for your feedback. This is a real-world study conducted at a single medical institution. We believe that the use of double immunohistochemistry with PD-L1 and CD163, combined with high-magnification microscopic analysis, provides valuable insight into the interactions between HRS cells and macrophages. When considered alongside EBV status—confirmed by molecular ISH—and clinical follow-up data, this approach has the potential to contribute meaningful new knowledge to the field. We analyzed ten fields per case, with an average of 40 HRS cells evaluated. This strategy ensured both precision and representativeness, resulting in cell counts comparable to those reported in other studies. Furthermore, applying the H-score to assess PD-L1 expression on HRS cells adds value, as PD-L1 intensity can vary significantly between individual cells—even within the same microscopic field. The H-score, by accounting for both intensity and proportion, allows for a more nuanced and accurate evaluation.
- Q: In Figure 4, the authors could have added arrows to more clearly indicate which cells are HRS cells and which are PD-L1-positive macrophages. Similarly, Figure 2 also lacks arrows, which would help in better identifying the macrophages.
A: Thank you for your suggestion, we have added the arrows and asterisk on both microphotographs.
- Given these significant limitations impacting the ability to draw definitive conclusions, the manuscript appears too weak in its current form.
A: Following the valuable suggestions from you and the other reviewers—for which we are truly grateful—we have revised the manuscript. We sincerely appreciate your input and hope that the updated version reflects those efforts.
Reviewer 4 Report
Comments and Suggestions for Authors
The study makes minor enhancements on the previous ones from Hollander et al. (Blood Adv. 2017;1(18):1427-1439). Authors must specify what makes their work more innovative compared to other research.
Minor comments:
1) Abbreviations in a manuscript should be explained upon their first use in the abstract and the main body of the text.
2) Methods (4.2.) should be divided into subsections. Evaluation of PD-L1 on HRS cells and macrophages should be described in separate subsections.
3) H-Score calculation should be explained.
4) Figure 4 should be moved to the Results section.
5) Definition of cutoff. Why not use ROC analysis?
6) Please make Table 3 more clearly presented and easier to read.
Author Response
- Q: The study makes minor enhancements on the previous ones from Hollander et al. (Blood Adv. 2017;1(18):1427-1439). Authors must specify what makes their work more innovative compared to other research.
A: The advantages of this study are the precise assessment od PD-L1 expression of HRS cells (H-score instead of percentage) and counting numbers and percentage of PD-L1+ macrophages. They were analysed on whole histological slides at high magnification, which allows insight into the fine details of the spacial relationship between HRS and macrophages.
We analysed ten fields with an average of 40 HRS cells evaluated per case. This approach maintained both precision and representativeness, resulting in cell counts comparable to those reported in other studies.
Applying the H-score to evaluate PD-L1 expression on HRS cells adds value. PD-L1 intensity can vary significantly from cell to cell—even within the same microscopic field—so using a scoring system that incorporates both intensity and proportion provides a more refined analysis.
Finally, unlike previous studies such as those by Koh et al. and Hollander et al., which assessed PD1 and PD-L1 expression across all leukocytes or the broader tumor microenvironment, we specifically distinguished between HRS cells and macrophages. Given that the prognostic significance of PD-L1 expression in cHL remains uncertain, we believe that ongoing, detailed PD/PD-L1 analysis in real-world clinical samples—like those collected over time in hospital centers—can contribute valuable insights into this disease.
- Q: Abbreviations in a manuscript should be explained upon their first use in the abstract and the main body of the text.
A: thank you very much, we corrected it.
- Methods (4.2.) should be divided into subsections. Evaluation of PD-L1 on HRS cells and macrophages should be described in separate subsections.
A: thank you for your suggestion, we divided them.
- H-Score calculation should be explained.
A: Thank you for your suggestion. We used a method similar to that of Roemer et al. and now described it in the Methods and cited in the References.
- Figure 4 should be moved to the Results section.
A: Thank you very much for your useful advice! We put the Figure 4 (now as a Figure 1) in the Results.
- Definition of cutoff. Why not use ROC analysis?
A: Given that expression was evaluated using the H-score, we applied the median value as a cutoff point to categorize cases into high and low expression groups. We did not use ROC analysis because our sample size is relatively small so dividing it into calibration and validation data sets would made it even smaller.
- Please make Table 3 more clearly presented and easier to read.
A: Thank you for your suggestion. We have followed your advice and revised it to make it simpler.
Round 2
Reviewer 2 Report
Comments and Suggestions for Authors
The authors addressed all queries
Reviewer 3 Report
Comments and Suggestions for Authors
The authors have made significant efforts to implement meaningful changes, including improving the clarity and accuracy of the manuscript’s text. Therefore, I accept the manuscript for publication in the journal IJMS.
The only issue is that the last figure is incorrectly referred to as Figure 3, while it should be Figure 4. This correction should also be made in the manuscript text wherever that figure is mentioned (lane 128-140).